# Challenges in Promoting Mitochondrial Transplantation Therapy

**DOI:** 10.3390/ijms21176365

**Published:** 2020-09-02

**Authors:** Yuma Yamada, Momo Ito, Manae Arai, Mitsue Hibino, Takao Tsujioka, Hideyoshi Harashima

**Affiliations:** 1Faculty of Pharmaceutical Sciences, Hokkaido University, Kita-12, Nishi-6, Kita-ku, Sapporo 060-0812, Japan; momopika@eis.hokudai.ac.jp (M.I.); mana_eknm1450@eis.hokudai.ac.jp (M.A.); bm0616@eis.hokudai.ac.jp (M.H.); harasima@pharm.hokudai.ac.jp (H.H.); 2Laboratory for Biological Drug Development based on DDS Technology, Hokkaido University, Kita-12, Nishi-6, Kita-ku, Sapporo 060-0812, Japan; 3Department of Pediatrics, Graduate School of Medicine, Hokkaido University, Kita-15, Nishi 7, Kita-ku, Sapporo 060-8638, Japan; takaotsujioka@gmail.com

**Keywords:** mitochondria, mitochondrial transplantation, immunological reaction, mitochondrial storage, drug delivery, MITO-Porter

## Abstract

Mitochondrial transplantation therapy is an innovative strategy for the treatment of mitochondrial dysfunction. The approach has been reported to be useful in the treatment of cardiac ischemic reperfusion injuries in human clinical trials and has also been shown to be useful in animal studies as a method for treating mitochondrial dysfunction in various tissues, including the heart, liver, lungs, and brain. On the other hand, there is no methodology for using preserved mitochondria. Research into the pharmaceutical formulation of mitochondria to promote mitochondrial transplantation therapy as the next step in treating many patients is urgently needed. In this review, we overview previous studies on the therapeutic effects of mitochondrial transplantation. We also discuss studies related to immune responses that occur during mitochondrial transplantation and methods for preserving mitochondria, which are key to their stability as medicines. Finally, we describe research related to mitochondrial targeting drug delivery systems (DDS) and discuss future perspectives of mitochondrial transplantation.

## 1. Introduction

Mitochondrial transplantation therapy has been reported to be useful for the treatment of cardiomyopathy in a human clinical study in 2017 [1], and it has attracted considerable interest as an innovative therapeutic strategy for the treatment of mitochondrial dysfunctions. The first clinical study of mitochondrial transplantation therapy was held in Boston children’s hospital, which was reported by McCully and coworkers [1] and had a great impact in the field of medicine. They performed the autologous mitochondrial transplantation for myocardial ischemia–reperfusion injury of pediatric patients who required extracorporeal membrane oxygenation (ECMO), and they recovered most of their cardiac function and successfully separated from ECMO support. The usefulness of mitochondrial transplantation as a method for treating mitochondrial injuries in various tissues, including the heart, liver, lungs, and brain has been reported in animal studies. In this review, we summarize previously reported studies on the therapeutic efficacy of mitochondrial transplantation.

It should be noted that there is no way to use mitochondria for transplantation except in the currently ongoing clinical studies. Drug development and research to broaden the use of mitochondrial transplantation therapy as the next step in treating a wider spectrum of patients is strongly required. There are a number of critical issues that need to be overcome, which explains why there are only a few reports of clinical studies and no commercially available mitochondrial drugs.

An important aspect involves the issue of immune response, which is a concern during mitochondrial transplantation. Although it appears to be safe for autologous mitochondrial transplantation, it would be more practical to use allogeneic mitochondria rather than autologous mitochondria as a pharmaceutical product. The reason is that there would be a more suitable situation for allogenic mitochondria than autologous, including the patients of mitochondrial disease whose cells and tissues have the mutation of mitochondrial DNA and mitochondrial dysfunction. Therefore, we summarize the current state of our knowledge regarding immune responses during mitochondrial transplantation, which is an important factor in mitochondrial transplantation therapy. We also discuss strategies for preserving mitochondria, which is the key to their stability as a pharmaceutical product. To further develop conventional mitochondrial transplantation, research on a drug delivery system (DDS) for delivering mitochondria to target tissues and into target cells is of prime importance. Research on DDS technology to target mitochondria contained in diseased cells would greatly accelerate the field of mitochondrial transplantation. 

## 2. Current Status of Mitochondrial Transplantation Therapy

In this section, reports of mitochondrial transplantation therapy and the related studies are described. The strategy for mitochondrial transplantation is to isolate and purify mitochondria from normal tissues or cells and then transplant them into the target tissue. In addition to the use of a commercially available kit for isolating mitochondria, the method for isolating mitochondria reported by McCully and coworkers [2] is frequently used. In this procedure, mitochondria are isolated by centrifugation of the collected tissue homogenates in a chilled buffer [2]. To obtain transplant mitochondria from cells, it has been reported that mitochondria can also be isolated from cell homogenates by sequential centrifugation [3,4].

The method used to isolate mitochondria differs depending on the mitochondrial source and the facility where mitochondrial isolation is performed. In addition, several protocols have been reported that include different conditions in terms of the target tissues and cells, target diseases, and routes of administration. Table 1, Table 2 and Table 3 summarize the disease models, target organs, sources of mitochondria, routes of administration, and therapeutic outcomes. Here, we mainly summarize reports of mitochondrial transplantation therapy for the treatment of heart diseases and the validation of this therapy using various animal models of such diseases.

### 2.1. Mitochondrial Transplantation Therapy for Heart Diseases

The therapeutic effect of isolated mitochondria was initially validated in myocardial-derived cells and in animal models of cardiac ischemia–reperfusion. In in vivo ischemia–reperfusion models, the isolated mitochondria were administered to the ischemic myocardium of the model immediately prior to reperfusion. Reports of the validation of mitochondria transplantation therapies for the treatment of heart diseases are summarized in Table 1. 

McCully and coworkers validated a mitochondrial transplantation therapeutic strategy using an ischemia–reperfusion model of the rabbit heart [5]. They isolated mitochondria derived from the left ventricle of the rabbit and administered the isolated mitochondria directly to the site of partial ischemia–reperfusion. This mitochondrial transplantation procedure resulted in a significant reduction in the levels of creatine kinase-MB (CK-MB) and cardiac Troponin I (cTnI), which are markers of myocardial infarction, and caspase-3, a marker of apoptosis. Moreover, the recovery of myocardial function was observed.

In addition, using a model similar to the ischemia–reperfusion model of the rabbit heart, the therapeutic effects of mitochondrial transplantation using mitochondria isolated from an autologous non-ischemic pectoralis major muscle as the source were validated. The results of these investigations provided confirmation that a significant reduction in the extent of infarction was observed, as indicated by the continued reduction in the ratio of the extent of the infarct to the area at risk (IS/AAR) up to 4 weeks after the transplantation [6]. 

The therapeutic effects of mitochondrial transplantation using autologous mitochondria isolated from the pectoralis major muscle in a porcine model of ischemia/reperfusion were validated using various protocols. In a study reported by Kaza et al. [7], mitochondria were administered immediately prior to reperfusion following cardiac ischemia, as was done in the case of the validation of mitochondrial transplantation using other animals. The results showed a decrease in IS/AAR, and the injected mitochondria were observed for periods of up to 4 weeks after administration by using magnetic resonance image (MRI) imaging. In a study reported by Guariento et al. [8], the effects of single or multiple intracoronary doses of isolated mitochondrial before ischemic manipulation were investigated for use as a preventive therapeutic effect. The results showed the recovery of myocardial function and a decrease in IS/AAR in the mitochondrial transplantation group. Blitzer et al. [9] examined the issue of whether isolated mitochondria could be administered to the coronary artery during reperfusion procedures, rather than immediately before reperfusion and concluded that this would provide a therapeutic benefit. The results showed a temporary recovery of coronary blood flow after mitochondrial transplantation.

In a rat model of diabetes, the therapeutic effect of mitochondrial transplantation was investigated by administering isolated mitochondria to the heart [10]. When the rat heart was subjected to warm global ischemia, autologous or allogeneic mitochondria isolated from the pectoralis major muscle was administered via the coronary artery. The recovery of left heart function was observed after the mitochondrial transplantation of both autologous and allogeneic isolated mitochondria.

### 2.2. Attempts Related to Mitochondrial Transplantation Targeting the Liver, Lungs, and Brain Using Animal Models

The transplantation of allogenic mitochondria was attempted using animal models of liver, lungs, and brain diseases (Table 2). Mitochondrial transplantation in the liver was attempted using a model rat with a liver ischemia–reperfusion injury [11]. The transplantation of allogeneic liver mitochondria via intrasplenic injection resulted in a reduction in the levels of alanine aminotransferase (ALT), the levels of apoptotic markers, and the production of reactive oxygen species (ROS) [11]. Using a mouse model of a fatty liver, the therapeutic effect was also validated by multiple intravenous injections of allogenic liver mitochondria at multiple 3-day intervals. The results indicated that the injection of isolated mitochondria resulted in a significant reduction in fatty liver endpoints such as ALT levels [12].

The transplantation of allogeneic mitochondria into the lungs of a mouse model of ischemia–reperfusion was investigated. The mitochondria were isolated from the gastrocnemius muscle and either directly injected into the pulmonary artery or via nebulization (aerosol delivery via trachea). The findings confirmed that the transplantation resulted in the levels of markers of pulmonary function, including dynamic compliance and resistance, to approach close to normal levels [13].

Mitochondrial transplantation to treat nervous system dysfunctions was investigated using rat models of a Spinal Cord Injury (SCI) [14]. The allogenic mitochondria were isolated from PC-12 cells, a cell line derived from the pheochromocytoma of the rat adrenal medulla, or rat soleus muscles, and the isolated mitochondria were then transplanted into the mediolateral gray matter of the rats. The findings indicated that the recovery of mitochondrial oxygen consumption rate (OCR) was dependent on the mitochondrial dose that was used. The administration of mitochondria could be one of the more useful therapeutic strategies for treating SCI, but the long-term efficacy including the measurement of OCR would need to be further investigated.

### 2.3. Evaluation of Cell Function after Mitochondrial Transplantation under Cell Culture Conditions

Several attempts to evaluate cell function after mitochondrial transplantation under cell culture conditions have been reported (Table 3). Co-culturing cardiomyocytes with allogeneic mitochondria derived from cardiomyocytes showed mitochondrial internalization and an increase in the levels of ATP production [15,16]. For the validation of a therapeutic strategy for treating diseased cells, it has been reported that co-culturing breast cancer cells with allogeneic mitochondria isolated from mesenchymal stem cell (MSCs) resulted in a dose-dependent restoration of mitochondrial OCR [17].

In addition, kidney proximal tubular epithelial cells (PTECs) from streptozotocin (STZ) induced diabetic model rats co-cultured with allogeneic MSCs-derived mitochondria reduced the level of apoptosis and ROS production [18]. In this study, NRK-52E cells, rat renal proximal tubular cells, were exposed to a high glucose environment co-cultured with isolated MSCs derived mitochondria or with isolated mitochondria derived from NIH-3T3 cells and mouse fibroblast-like cells, respectively. As a result, it was confirmed that mitochondria derived from NIH-3T3 failed to decrease the levels of ROS production, although isolated MSC-derived mitochondria decreased the levels. These results suggest that the origin of the isolated mitochondria might be a factor in the therapeutic effect such as the suppression of ROS production.

### 2.4. The Only Clinical Trial and Issues to Overcome in Mitochondrial Transplantation Therapy

Mitochondrial transplantation therapy by administering isolated mitochondria to a diseased site would be promising, including a clinical trial related to the treatment of the human heart, as well as the in vitro and in vivo studies described above. In the single clinical trial by McCully and coworkers [1] that had a deep impact in the medical world, pediatric patients with severe heart disease who were difficult to separate from ECMO due to myocardial ischemia–reperfusion injury were selected in this study. Autologous rectus abdominis muscle tissue was harvested and mitochondria were isolated and administered to myocardial sites with poor contractile movement, resulting in recovery from the need for ECMO support in 4 of 5 patients.

McCully et al. presumed that the mechanism by which transplanted mitochondria exert a protective effect on the ischemic heart is that the mitochondria promote an enhanced myocardial function, as described below [19]. One possibility is that transplanted mitochondria increase the ATP content and activate ATP synthesis in the cells of the heart. Another possibility is that transplanted mitochondria migrate into the heart cells by actin-dependent endocytosis, and the mitochondria then release cardioprotective cytokines. These cytokines could promote cell growth and proliferation, thus promoting angiogenesis and protecting cardiomyocytes from apoptosis. The third possibility is that transplanted mitochondria replace damaged mitochondrial DNA (mtDNA) with normal mtDNA during ischemia. Replacing damaged mtDNA with normal mtDNA by mitochondrial transplantation could increase ATP synthesis capacity. Based on these possibilities, it is suggested that mitochondrial transplantation could increase myocardial function.

It should be noted that there are several issues that need to be overcome for improving mitochondrial transplantation therapy. For example, a single administration of mitochondria does not result in the maintenance of long-term therapeutic efficacy as described above. In the current situation, the method of mitochondrial isolation, the mitochondrial source, the route of administration, and number of doses are all dependent on the ease of performing mitochondrial transplantation. It is expected that optimal standard protocols for mitochondrial transplantation will need to be developed for each target disease.

## 3. Immunological Reaction of Transplanted Mitochondria

For mitochondrial dysfunctions due to acquired injuries such as ischemia and physical damage, mitochondria isolated from the intact tissue of the same patient should be transplanted. On the other hand, in various mitochondrial diseases that are caused by a congenital mitochondrial dysfunction, the transplantation of autologous mitochondria would not be suitable, because there is some probability that mitochondria in all tissues could be dysfunctional; thus, allogeneic mitochondrial transplantation would be required. Therefore, as with various organ transplants, the immune response to allogenic mitochondria should be well discussed. Research reports of the immune response for mitochondrial transplantation are summarized in Table 4.

Immune responses in mitochondrial transplantation are a subject that has not been adequately discussed, except for studies of an immune response after mitochondrial transplantation into the myocardium reported by McCully and coworkers. Masuzawa et al. reported on the usefulness of a single transplantation of isolated autologous mitochondria derived from pectoral major muscle tissue in a rabbit model of ischemic cardiomyopathy. The immune responses to this autograft were also investigated, and no significant increase in various inflammatory markers in serum and no anti-mitochondrial antibodies were detected after the mitochondrial transplantation [6]. Cytokine and chemokine activity from human peripheral blood mononuclear cells against mitochondria isolated from HeLa cells were also examined, and there was no detectable increase in cytokine activity related to rejection after heart transplantation.

Kaza et al. reported on an immune response after a single autologous mitochondrial transplantation in a porcine model of ischemia/reperfusion [7]. In this study, serum cytokine levels were measured as an index of the inflammatory and immune response after the transplantation, but no significant increase was observed.

Ramirez-Barbieri et al. investigated the immune response and damage-associated molecular patterns (DAMPs) responses after mice were intraperitoneally injected with single or multiple doses of allogenic mitochondria [20]. The results indicated that there was no increase in cytokine and chemokine levels, including interleukin-2 (IL-2), interferon-γ (IFN-γ), or immunoglobulin M (IgM) in response to the transplanted mitochondria, either in serum recovered after the injections of autologous and allogenic mitochondria. Furthermore, no increase in mtDNA levels in the blood that could potentially cause tissue damage due to DAMPs was detected, nor was any tissue damage detected in the lung or heart tissue.

On the other hand, several studies have concluded that there is an immune response after mitochondrial transplantation. Pollara et al. reported that mitochondria-derived DAMPs were abundant in the blood of postmortem organ transplant donors, and that pro-inflammatory cytokines and chemokines are concomitantly elevated [22]. It was also reported that the amount of mtDNA was correlated with early allograft dysfunction in liver transplantation patients. Lin et al. reported that extracellular mitochondria activated vascular endothelial cells resulted in the production of inflammatory cytokines and chemokines [21]. They also showed that the administration of isolated mitochondria significantly increased rejection in a mouse model of a xenograft heart transplant [21].

A few reports have appeared concerning immune responses during mitochondrial transplantation to date. Understanding the mechanism responsible for an immune response during mitochondrial transplantation would be of value in terms of reducing the risk associated with mitochondrial transplantation. It would also be very interesting from a scientific standpoint, and further research will likely be carried out in the future.

## 4. Storage of Mitochondria

In a clinical study of mitochondrial transplantation in humans, the protocol involves isolating mitochondria from the patient’s rectus abdominis muscle and transplanting them into the diseased site during mitochondrial transplantation surgery. Since the isolated mitochondria are significantly less active when stored on ice for more than 1 h [19], a rapid operation is essential. If mitochondria for transplantation can be used, not as a preparation for each use, but as a storable preparation for mitochondrial transplantation, clinical applications of mitochondrial transplantation would be significantly advanced. To this end, the establishment of a method that permits mitochondria to be stored for an extended period of time is a very important issue.

When mitochondria are stored in a refrigerated or frozen state, damage to the outer and inner mitochondrial membranes occurs. When the mitochondrial inner membrane is impaired, there is a decrease in energy production due to oxidative phosphorylation, ATP synthesis capacity, the mitochondrial inner-membrane transport of certain molecules, and the biomolecular synthesis such as amino acids, along with a decrease in the pH gradient and membrane potential. When the mitochondrial outer membrane is impaired, it loses its integrity and membrane permeability is increased, resulting in the release of proteins, including cytochrome c (cyt c), from the mitochondrial intermembrane space. As a result, apoptosis is induced due to caspase activation, regardless of Bcl family protein responsiveness. Therefore, when the mitochondrial outer membrane is damaged, apoptosis proceeds in an uncontrolled manner [23].

Thus, mitochondrial storage under conditions where the stability of the mitochondrial outer and inner membranes is maintained is needed, in order for mitochondrial function to be maintained during mitochondrial refrigeration and freezing. In order to maintain mitochondrial activity and long-term storage, it is necessary to maintain mitochondrial structure. In this section, we discuss investigations related to the cold storage and cryopreservation of mitochondria, which was carried out with the aim of maintaining the stability of isolated mitochondria during long-term storage.

### 4.1. Cold Storage of Mitochondria

McCully and coworkers reported that the activity of isolated mitochondria is greatly reduced when they are stored on ice for more than 1 h [19]. The composition of the mitochondrial storage solution is also an important factor that contributes to the maintenance of mitochondrial activity when they are stored under cold conditions. Mitochondrial activities and structure are described below, in the case of the storage of mitochondria in cold 4-(2-HydroxyEthyl)-1-PiperazineEthaneSulfonic acid (HEPES)–sucrose-based buffer, the University of Wisconsin (UW) solution, and the Eurocollins solution.

Gnaiger E et al. reported on mitochondrial function when the mitochondria were stored in a HEPES–sucrose-based buffer on ice [24]. The HEPES–sucrose-based buffer contained antioxidants, ATP, histidine, and colloidal agents, which served as a preservation buffer. They evaluated the extent to which mitochondria isolated from rat hearts continue to show respiratory capacity, when they are stored on ice in preservation buffer. After isolated mitochondria in preservation buffer were refrigerated for 24 h, the respiratory capacity was maintained for more than 80%. Moreover, the addition of cyt c to the preservation buffer permitted the respiratory capacity to be maintained at approximately 100% for 24 h after the start of refrigeration.

On the other hand, storing mitochondria for more than 2 days under the above conditions resulted in a decrease in respiratory capacity due to the impairment of several mitochondrial functions. Cyt c release and the electron transport of complex II were the most heavily affected by the impairment of the refrigerated mitochondria. In summary, the use of a preservation buffer permits approximately 80% of mitochondrial respiratory capacity to be maintained after one day (24 h) of mitochondrial refrigeration, while cyt c was released from the isolated mitochondria.

The evaluation of the function and structure of mitochondrial isolated from rat liver that were stored in Eurocollins and UW solutions at 4 °C [25] is summarized below. Both the Eurocollins solution [26] and the UW solution [27] were developed for the purpose of preserving organs in the 1980s. Although both solutions are mainly phosphate buffered, the UW solution contains antioxidants such as allopurinol, adenosine, and glutathione. The UW solution could maintain the cyt c content and complex II activity of the mitochondria isolated from rat liver after storage for 24 hon ice. The maintenance of the cyt c content can be attributed to the presence of magnesium as a membrane stabilizer and a colloid that prevents cell and mitochondrial expansion. It was confirmed that the activity of complexes III and IV in mitochondria isolated from rat liver that had been stored for 7 hon ice could be maintained, and this can be attributed to the inhibition of ROS production by the antioxidants in the solution. On the other hand, when rat livers were stored in the Eurocollins solution, glucose permeated into the hepatocytes, mitochondrial expansion was observed, and complex III and IV activities were lost due to the absence of antioxidants.

These results indicate that the storage of mitochondria under conditions of refrigeration in all of the preservation buffers, UW solution, and Eurocollins solution have limited ability to protect mitochondria from damage. Thus, the optimizing of the preservation solution used in mitochondrial cold storage is needed to maintain mitochondrial structure and function. The fact that the addition of antioxidants was effective in maintaining complex III and IV activity and the addition of colloids was effective in maintaining the mitochondrial structure could be useful for this optimization.

### 4.2. Cryopreservation of Mitochondria

Reports of the cryopreservation of mitochondria using a preservation solution have been reported (Table 5). Greiff et al. stored mitochondria that had been frozen in dimethyl sulfoxide (DMSO) or glycerol and investigated mitochondria function [28]. As a result, the oxidative phosphorylation capacity of the frozen mitochondria remained unchanged compared to the non-frozen mitochondria in this experiment, where mitochondria were stored in 10% of DMSO at −65 °C for 18 days or 10% of glycerol at −65 °C for 15 days. 

Nukala et al. evaluated mitochondrial activity, after the mitochondria isolated from a rat cerebral cortex had been stored frozen in a 10% DMSO preservation solution at −80 °C for 1 week [29]. Electron microscopic observations indicated that the inner and outer mitochondrial membranes were intact. These results indicate that a preservation solution that contains DMSO permits the oxidative phosphorylation capacity to be retained. On the other hand, cyt c retention in mitochondria that were frozen in 10% DMSO was reduced compared to normal mitochondria, although the extent of release of the cyt c could be reduced compared to mitochondria that were frozen in the absence of DMSO.

Mitochondrial toxicity caused by DMSO was investigated when the isolated mitochondrial were stored in 30% DMSO in a frozen status. In the case where frozen isolated mitochondria were stored in 30% glycerol, the mitochondrial oxidative phosphorylation capacity of mitochondria was inactivated, whereas the capacity of mitochondria stored in 30% DMSO was retained, indicating that DMSO is less toxic than glycerol [28]. However, there was a decrease in mitochondrial activity when DMSO was used to store the mitochondria compared to normal mitochondrial activity [29], which is thought to be due to the toxic effect of DMSO on mitochondrial energy production [29]. In summary, the advantage of DMSO is that it prevents mitochondrial membrane damage upon freezing and thawing, but the disadvantage is that it does not allow a membrane structure capable of retaining cyt c to be maintained, and accordingly, its energy-related functions are not retained after long-term storage.

Yamaguchi et al. examined the protective effect of Trehalose on mitochondrial cryopreservation [30]. In terms of cyt c retention, freezing and thawing in a sucrose/mannitol buffer released 95% of the cyt c within 15 min after thawing, whereas when Trehalose was used, it was possible to maintain approximately 95% of the cyt c within 45 min after thawing. In addition, it was confirmed that the apoptosis-inducing capacity of mitochondria cryopreserved in Trehalose was unchanged. On the other hand, ATP synthesis and mitochondrial respiratory capacity were reduced to about 30% of the normal level, indicating that the energy production capacity of mitochondria in the case of the samples frozen with Trehalose was maintained. The advantage of Trehalose is that the regulation of mitochondrial apoptosis can be maintained, while the disadvantage is that bioenergetic production capacity is greatly reduced compared to normal conditions.

The above findings indicate that it is not possible to maintain the mitochondrial energy production capacity at the same level as occurs under normal conditions when cryopreservation is being used, but it is possible to maintain mitochondrial membrane stability. Therefore, there is clearly a need to develop a storage solution in which frozen mitochondria can maintain their normal bioenergetic production capacity.

## 5. Drug Delivery Systems to Accelerate Mitochondrial Transplantation

### 5.1. Attempts to Achieve the Cellular Uptake of Mitochondria by Using a Cell-Penetrating Peptide

A few studies dealing with the delivery of mitochondria to the target tissues and cells to improve mitochondrial transplantation have appeared. Currently, mitochondrial transplantation therapies have been investigated for use in acute diseases and limited target sites, but the use of DDS technology could greatly expand the scope of mitochondrial transplantation-based therapies. To date, studies have been conducted to modify isolated mitochondria with a cell-penetrating peptide (CPP) to improve the efficiency of cellular uptake (Figure 1). Mitochondria have been reported to be taken up into cells by macropinocytosis [3,31], but the extent of the uptake is limited. Attempts have been reported to modify donor mitochondria with CPPs such as transactivator of transcription (TAT) peptide and Pep-1 and mitochondriotropic compounds, such as the triphenylphosphonium cation (TPP), which improve mitochondrial function [32,33]. In this section, reports of peptide-mediated mitochondrial delivery (PMD) using Pep-1-modified mitochondria (Pep-1-Mito) are mainly summarized.

Pep-1, a member of the CPP family of peptides, is an amphiphilic peptide that is composed of three domains: a hydrophobic site (tryptophan), a hydrophilic site (lysine), and a spacer region. It was reported that the mechanism responsible for cellular uptake by Pep-1 is not dependent on the endosomal pathway, but, rather, it involves membrane fusion followed by an electrostatic hydrophobic interaction with the cell membrane [34]. Chang et al. reported on various therapeutic effects of Pep-1-conjugated isolated mitochondria (Pep-1-Mito) using patient-derived dermal fibroblasts.

Co-culturing fibroblasts derived from the patient with myoclonic epilepsy with ragged red fibers (MERRF) with Pep-1-Mito resulted in the restoration of mitochondrial function starting at 3 days after the co-culture, and the function was maintained for at least 21 days. Compared to non-treated MERRF fibroblasts, the mitochondrial morphology was elongated, ROS production was reduced by 16%, and the mitochondrial membrane potential was increased by 2.3-fold in the case of MERRF fibroblasts with Pep-1-Mito [35]. Co-culturing fibroblasts derived from a patient with mitochondrial myopathy, encephalopathy, lactic acidosis, and stroke-like episodes (MELAS) with Pep-1-Mito showed that the Pep-1-Mito internalized into the fibroblasts and reached the mitochondria at 2 h after the start of the co-culture. In addition, mitochondrial respiration measurements showed a significant increase in basal respiration, ATP-linked OCR, and maximal respiration in the group co-cultured with Pep-1-Mito compared to the untreated group. It was concluded that this improvement in function is due to enhanced mitochondrial biogenesis and the normalization of fusion and fission of mitochondria [36].

In the nervous system, local injections of autologous/allogeneic Pep-1-Mito into the medial forebrain bundle (MFB) of a rat model of Parkinson’s disease were examined. The results revealed that motor performance was restored and the imbalance in the expression of respiratory chain complex proteins in the substantia nigra was improved [37].

In oncology, the co-culturing of Pep-1-Mito containing wild-type mtDNA with a breast cancer cell line induced cellular apoptosis due to the increase in the nuclear transfer of the apoptosis-inducing factor (AIF), which reduced cell proliferation and the level of oxidative stress. On the other hand, the co-culturing of Pep-1-Mito containing mutant mtDNA inhibited mitochondrial respiration and increased glycolytic activity [38]. Furthermore, the intratumor administration of Pep-1-Mito once per week for 4 weeks in a mouse model with triple-negative breast cancer showed an antitumor effect. These results were accompanied by the improvement of mitochondrial homeostasis in the tumor microenvironment by the introduction of normal mitochondria, which decreased the levels of oxidative stress, reduced the population of cancer-associated fibroblasts (CAF), and enhanced the infiltration of cytotoxic T cells [39].

The PMD treatment has the potential to expand mitochondrial transplantation and address challenges such as improving introduction efficiency, the selective targeting of the transplant mitochondria, and long-term stability of the mitochondrial formulation.

### 5.2. Mitochondrial DDS Technology

Attempts are currently underway to deliver therapeutic molecules to mitochondria by treating the mitochondria themselves in diseased cells. Mitochondria targeting DDS technology has the potential to make a significant contribution to the treatment of diseases related to mitochondrial dysfunction. Mitochondrial DDS targeting diseased cells might contribute to the acceleration of mitochondrial transplantation. In order to deliver therapeutic molecules to the mitochondria of a target cell, they must be taken up by the target cell, reach the mitochondria, become localized inside the mitochondria, and function inside the mitochondria once they are there. Candidate molecules range from small molecules such as antioxidant molecules to macromolecules such as proteins and genes. If therapies targeting mitochondria could be realized, an extremely wide variety of therapies would be possible, including cancer, ischemic disease, and gene therapies. Therefore, the ability to deliver a variety of molecules to the interior of mitochondria is one of the important factors in developing mitochondrial DDS. Here, we focus on our research on mitochondrial DDS.

Over the last decade, our focus has been on the development of a versatile mitochondrial targeting liposomal-based nanodevice, namely a MITO-Porter system [40,41,42,43,44]. Focusing on the fact that mitochondria undergo active fusion and fission within cells and share biomolecules such as nucleic acids and proteins with each other, we designed a MITO-Porter that has the ability to fuse with mitochondria for mitochondrial delivery. As shown in Figure 2, the MITO-Porter, the surface of which is modified with the positively charged cell penetrating peptide, octaarginine (R8), is taken up by the cell via macropinocytosis (1st step), binds to mitochondria with negative membrane potentials via electrostatic interactions (2nd step), and then delivers the cargoes to the mitochondria via membrane fusion (3rd step).

In this strategy, we concluded that the MITO-Porter represents a mitochondrial targeting DDS that does not limit the physical properties or size of the cargoes, because the MITO-Porter delivers the internally enclosed molecules (cargoes) to mitochondria via membrane fusion. We have continued our research on the development of nanomedicines based on MITO-Porter technology. This research includes fields such as mitochondrial gene therapy [45,46,47], cancer therapy [48,49,50,51,52], ischemic diseases therapy [42,53], and cell therapy [54].

In this section, we focus on our research related to mitochondrial targeting DDS. We are involved in a research project for “Drug discovery targeting mitochondria”. As part of this activity, BioVenture, LUCA Science Co., Ltd. was established on December 25 2018 based on technologies focusing on mitochondria including the MITO-Porter system. The goal of this effort is to promote research on the development of mitochondrial nanomedicines, in which we served as scientific advisors. In addition, Hokkaido University’s Industry Creation Laboratories (Laboratory for Biological Drug Development base on DDS Technology) was opened in April 2020, and this has further accelerated the research and development of “Nano-medicines”.

## 6. Conclusions

In this review, we provided overviews of previous reports regarding mitochondrial transplantation. Mitochondrial transplantation therapy, which involves the use of mitochondria as therapeutic molecules appears to have a very promising future. The pharmaceutical formulation of mitochondria promises to be an important process for the adaptation of this therapy to a large number of patients. We hope that this review will stimulate and also accelerate research related to the development of innovative medicines based on mitochondrial transplantation.

## Figures and Tables

**Figure 1 ijms-21-06365-f001:**
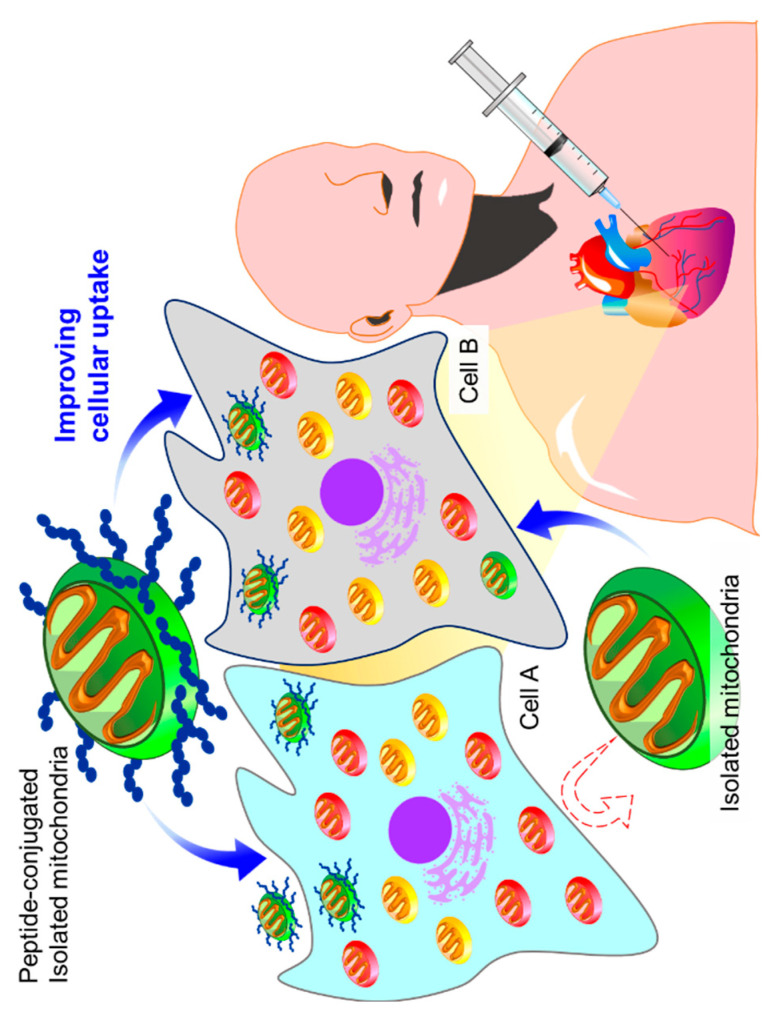
Strategy to improve the cellular uptake of isolated mitochondria. Cellular uptake capacity of isolated mitochondria is low and not taken up by a certain cell. Peptides such as a cell-penetrating peptide improve the cellular uptake of the isolated mitochondria.

**Figure 2 ijms-21-06365-f002:**
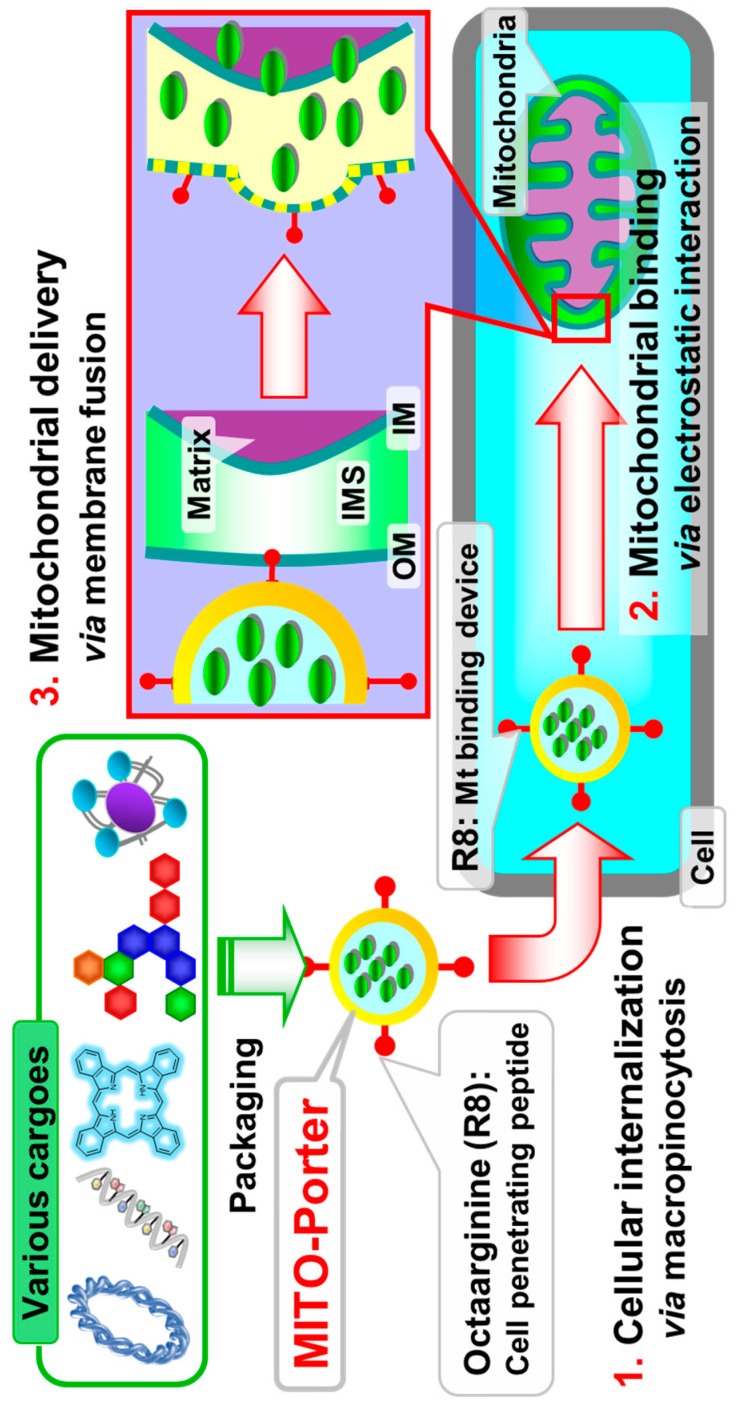
Schematic image of the use of a MITO-Porter for mitochondrial delivery. The MITO-Porter is efficiently internalized via macropinocytosis as the initial step. The second step is the interaction of the MITO-Porter with the mitochondrial membrane. Finally, the cargoes are delivered into mitochondria via membrane fusion. IM, inner membrane; IMS, intermembrane space; outer membrane.

**Table 1 ijms-21-06365-t001:** Research reports of mitochondrial transplantation in the heart.

Disease Models	Targeted Organs	Mitochondrial Sources	Injection Sites and Methods	Therapeutic Outcomes	References
New Zealand white rabbits with ischemia-reperfusion heart	Heart	Mt isolated from left ventricular of rabbit (allogeneic) by homogenization and centrifugation	Direct injection into RI zone of the heart	Reduction of CK-MB, caspase-3 activity, IS/AAR. Recovery of myocardial function	McCully et al., 2009 [5]
New Zealand white rabbits with ischemia-reperfusion heart	Heart	Mt isolated from pectoralis major muscle tissues of rabbit (autologous) by McCully’s method [5]	Direct injection into RI zone of the heart	Reduction of IS/AAR (after 2 h and 28 days of recovery), CK-MB, cTnI, and TUNEL positive cell nuclei. Increase of ATP content	Masuzawa et al., 2013 [6]
Yorkshire pigs with ischemia-reperfusion heart	Heart	Mt isolated from pectoralis major muscle tissues (autologous) by McCully’s method [2]	Direct injection into AAR zone of the heart	Reduction of markers of myocardial injury (3 days), IS/AAR. Presence of mitochondria (4 weeks after injection in pig heart)	Kaza et al., 2017 [7]
Yorkshire pigs with ischemia-reperfusion heart	Heart	Mt isolated from pectoralis major muscle tissues (autologous) by McCully’s method [2]	Intracoronary injection (single bolus/serially)	Increase in CBF during the pre-RI period and throughout reperfusion, systolic function recovery, reduction of IS/AAR	Guariento et al., 2019 [8]
Yorkshire pigs with ischemia-reperfusion heart	Heart	Mt isolated from pectoralis major muscle tissues (autologous) by McCully’s method [2]	Injection into the left coronary ostium	Myocardial function recovery, increase in CBF (evident 15 min after injection)	Blitzer et al., 2020 [9]
Zucker Fatty rats with ischemia-reperfusion heart	Heart (with diabetics)	Mt isolated from pectoralis major muscle tissues (autologous/allogeneic) by McCully’s method [2]	Delivery to the coronary arteries via the aortic cannula	Left ventricular function recovery, reduction of IS/AAR	Doulamis et al., 2020 [10]
Pediatric patients with ischemia-reperfusion associated myocardial function	Heart	Mt isolated from rectus abdominis muscle tissues(autologous) by Masuzawa’s method [6]	Direct injection into RI zone of the heart	4 out of 5 patients successfully separated from ECMO support	Emani et al., 2017 [1]

CBF, coronary blood flow; CK-MB, creatine kinase-MB; cTnI, cardiac troponin; ECMO, extracorporeal membrane oxygenation; IS/AAR, infarct size/area at risk; Mt, mitochondria; RI, regional ischemia.

**Table 2 ijms-21-06365-t002:** Research reports of mitochondrial transplantation in the liver, lung, and brain.

Disease Models	Targeted Organs	Mitochondrial Sources	Injection Sites and Methods	Therapeutic Outcomes	References
Wistar rats with ischemia-reperfusion liver	Liver	Mt isolated from left ventricular of rabbit (allogeneic) by homogenization and centrifugation	Injection into spleen	Reduction of level of ALT, TUNEL-positive cells, markers of apoptotic pathways and ROS production	Lin HC et al., 2013 [11]
C57BL/6J mice with fatty liver	Liver	Mt isolated from HepG2 cells (xenogeneic) by using mitochondrial isolation and purification kit	Injection intravenously	Decrease level of ALT, AST, TC and LDL-C. Reduction of ROS production. Increase of ATP contents	Fu et al., 2017 [12]
C57BL/6J mice with ischemia-reperfusion lung	Lung	Mt isolated from gastrocnemius muscle of mice (allogeneic) by McCully’s method [2]	Direct injection into pulmonary artery/nebulization (aerosol delivery via trachea)	Decrease of resistance. Increase in dynamic compliance and inspiratory capacity	Moskowizova et al., 2019 [13]
Sprague-Dawley rats with SCI	Brain	Mt isolated from PC-12 cells (allogeneic) and soleus muscle of rat (allogeneic) by Gollihue’s method [4]	Injection into the mediolateral gray matter	Maintenance of OCR	Gollihue et al., 2018 [14]

ALT, alanine aminotransferase; LDL-C, low density lipoprotein cholesterol; Mt, mitochondria; OCR, oxygen consumption rate; RI, regional ischemia; ROS, reactive oxygen species; SCI, spinal cord injury; TC, content of cholesterol; TUNEL, terminal deoxynucleotidyl transferase dUTP nick end labeling.

**Table 3 ijms-21-06365-t003:** Research reports regarding mitochondrial transplantation under cell culture conditions.

Target Cells	Mitochondrial Sources	Therapeutic Outcomes	References
Cardiomyocytes of rat	Mt isolated from liver of rat (allogeneic) by Masuzawa’s method [6]	Mitochondrial uptake in a time-dependent manner, increase of ATP contents	Pacak et al., 2015 [15]
HeLa ρ_0_ cells	Mt isolated from HeLa cells (allogeneic) by Masuzawa’s method [6]	Increase in ATP contents (sustained until 3 weeks) and oxygen consumption rate	Pacak et al., 2015 [15]
induced pluripotent stem (iPS) cardiomyocytes of human	Mt isolated from cardiac fibroblasts of human (allogeneic) by replacing with buffer, filtration, and centrifugation	Co-localization between endogenous and exogenous mitochondria, increase in ATP contents in a time-dependent manner	Cowan et al., 2017 [16]
MDA-MB-231 cells (human breast cancer)	Mt isolated from MSCs of human (allogeneic) by using mitochondria isolation kit for cultured cells	Increase in OCR in a dose-dependent manner	Caicedo et al., 2015 [17]
Renal PTECs (diabetic neuropathy)	Mt isolated from MSCs of rats (allogeneic) by Kitani’s method [2]	Reduction of shrunken nuclei, ROS production, and TUNEL-positive apoptotic cells	Konari et al., 2019 [18]

MSCs, mesenchymal stem cells; Mt, mitochondria; OCR, oxygen consumption rate; PTECs, proximal tubular epithelial cells; RI, regional ischemia; ROS, reactive oxygen species; TUNEL, terminal deoxynucleotidyl transferase dUTP nick end labeling.

**Table 4 ijms-21-06365-t004:** Research reports of the immune response to mitochondrial transplantation.

Source	Methods and Number of Times of Administration	Recipients	Immune Response	References
Autologous pectoralis major muscle	8 × 0.1 mL direct injections of sterile respiration buffer containing mitochondria 9.7 × 10^6^ ± 1.7 × 10^6^/mL into the area at risk, only one time	In vivo: Male New Zealand white rabbits In vitro: Human peripheral blood mononuclear cells	In vivo: No increase of sensitive serum inflammatory markers. No detection of serum anti-mitochondria antibody after transplantation In vivo: No upregulation of cytokines and chemokines associated to acute heart transplantation rejection against mitochondria from HeLa cells	Masuzawa et al., 2013 [6]
Autologous pectoralis major muscle	8 × 0.1 mL injections of sterile respiration buffer containing mitochondria (9.9 × 10^7^ ± 1.4 × 10^7^/mL; 1.3 × 10^7^ mitochondria per injection site) directly into the area at risk, only one time	Female Yorkshire pigs	No significant difference of immune and inflammatory response and cytokine activation after 4 weeks of recovery	Kaza et al., 2017 [7]
Gastrocnemius muscle and quadriceps femoris muscle from syngeneic or allogeneic mice	Single i.p. injection of either syngeneic or allogeneic mitochondria at the concentration of 1 × 10^5^, 1 × 10^6^ or 1 × 10^7^.	Female BALB/cJ and C57BL/6J mice, age 5–8 weeks	No significant difference between control and mice receiving mitochondira injections in:	Ramirez-Barbieri, et al., 2019 [20]
			IL-2 and IFN-γ Serum IgM levels	
			Mean graft survival time	
			Lymphocyte infiltration and fibrosis in pathological analyses of donated skins	
			Circulating mitochondria DNA by real-time PCR	
	Serial i.p. injections of either syngeneic or allogeneic mitochondria at a concentration of 1 × 10^7^		Histopathological findings of lung and heart in HE stain and electron microscopy	
Mice LMTK cell line	In vivo:	In vivo: C57BL/6 mice	In vivo: Significantly earlier rejection of cardiac allografts treated with mitochondria Higher ISHLT grades of acute rejection in mitochondria-treated cardiac allografts	Lin, et al., 2019 [21]
	Intravenous injection of 300 μg of allogeneic mitochondria to BALB/c mice once on the day before harvest of the heart			
	In vivo	In vivo: Mice bEnd.3 cell line	In vitro: Activation of ECs, production of inflammatory cytokines and chemokines by co-incubation of mice ECs with allogenic mitochondria	
	Co-incubation of mice bEnd.3 cells with 50 μg/mL of allogeneic mitochondria for 5 h			
Human HeLa cell line	In vitro:	In vitro: HAECs	In vitro:	Lin, et al., 2019 [21]
	Co-incubation of confluent HAECs with 100 μg/mL of allogeneic mitochondria for 6 h		Activation of ECs, production of inflammatory cytokines and chemokines by co-incubation of HAECs with allogenic mitochondria	

DC, dendritic cell; EC, endothelial cell; HAEC, human aortic endothelial cell; HE, hematoxylin and eosin; IFN-γ, interferon-gamma; IgM, immunoglobulin M; IL-2, interleukin-2; i.p., intraperitoneal; ISHLT, International Society for Heart and Lung Transplantation.

**Table 5 ijms-21-06365-t005:** Summary of the cryopreservation of mitochondria.

Mitochondrial Source	Preservation Condition	Storage Solution	Mitochondrial Condition	References
Mitochondria isolated from rat livers	−65 °C frozen, in 18 days	10%, 30% DMSO	Oxidative phosphorylation is preserved.	Greiff et al., 1961 [28]
	−65 °C frozen, in 15 days	10%, 30% Glycerol	Oxidative phosphorylation is preserved in the case of 10% glycerol.	
Mitochondria isolated from rat cerebral cortex	−80 °C frozen, in a week	10% DMSO	Mitochondrial membranes and cristae structure are preserved. Mitochondrial respiratory capacity is reduced (80% compared to non-frozen condition).	V. Nukala et al., 2006 [29]
Mitochondria isolated from mouse liver	Frozen preservation	Trehalose	Oxidative phosphorylation and ATP production are reduced. Apoptosis inducibility, outer and inner membranes are preserved.	Yamaguchi et al., 2007 [30]
		Mannitol/sucrose	Cyt c content is lost. Mitochondrial swelling and cristae degradation are observed.	

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
