# Peer review of "Challenges in Promoting Mitochondrial Transplantation Therapy"

_ijms, 2020, doi:10.3390/ijms21176365_

Round 1

Reviewer 1 Report

In this manuscript Yamada et al reviewed therapeutic effects of and immune responses to mitochondrial transplantation, methods for preserving mitochondria and mitochondrial drug delivery systems (DDS). They focused on mitochondrial transplantation therapy for the treatment of heart diseases, as well as liver, lungs and brain diseases in animal models. Apart from the need for the extensive English language editing, here are my comments and suggestions:

Introductory chapter is basically repeated and somewhat expanded Abstract and as such should be re-written in order to better place this study in a broader context and highlight its importance.

Lines 51-55: drug delivery systems (DDS) to introduce mitochondria into target tissues/cells is not the same as DDS to target mitochondria in diseased cells.

Line 117 – “the activity of mitochondrial markers” is not a correct formulation because it is not their activity that was measured and should be corrected.

Line 133 – please explain “mechanical threshold”.

In the Chapter 2.4 only one clinical trial is mentioned, despite the title. Lines 157-9 are repetitive (written already in the lines 33-5).

Line 178 – the sentence is not finished.

Line 210 – the reference is missing.

Line 221 – this sentence needs re-phrasing, “utility preparation” is a strange term.

Line 229 – reference 23 is wrong and should be corrected.

Line 253 – “the decrease in respiratory capacity was inhibited” should be re-written.

Line 260 – “refrigerated – impairment” should be changed.

Lines 293-5 – after the “although the release of cyt c…” is not clear, please explain better.

Line 337 – What does “Culler” stand for?

Line 349 – ROS production?

Line 350 – replace “activated” with “increased”.

Line 366 – delete “system” and please explain “cellular activation”.

Line 384 – “There are an extremely wide variety of therapies targeting mitochondria is in contrast with what is said in the Line 43: “there are… no commercially available mitochondrial drugs”.

Line 394 – Add “membrane” before “potentials”.

Lines 409-16 – Ongoing projects and authors’ activities in academia and industry should not be a part of the scientific paper.

Lines 418 – 420 are repetitive and should be deleted.

Line 422 – please explain what does “The formulation of mitochondria” stand for.

Line 597 – The title of the Table 3 is wrong.

Author Response

To reviewer #1

       We are grateful to reviewer #1 for the critical comments and useful suggestions that have helped us to improve our paper. As indicated in the responses that follow, we have taken all these comments and suggestions into account in the revised version of our paper.

Comment:

In this manuscript Yamada et al reviewed therapeutic effects of and immune responses to mitochondrial transplantation, methods for preserving mitochondria and mitochondrial drug delivery systems (DDS). They focused on mitochondrial transplantation therapy for the treatment of heart diseases, as well as liver, lungs and brain diseases in animal models. Apart from the need for the extensive English language editing, here are my comments and suggestions:

Introductory chapter is basically repeated and somewhat expanded Abstract and as such should be re-written in order to better place this study in a broader context and highlight its importance.

We attempted to revise the introductory chapter in the revised version.

Lines 51-55: drug delivery systems (DDS) to introduce mitochondria into target tissues/cells is not the same as DDS to target mitochondria in diseased cells.

As the reviewer #1 pointed out, DDS to introduce mitochondria into target tissues/cells is not the same as DDS to target mitochondria in diseased cells. However, we would like to include mitochondrial DDS as we thought it was a important section.

Line 117 – “the activity of mitochondrial markers” is not a correct formulation because it is not their activity that was measured and should be corrected.

Thank you very much for the useful comment. We revised this sentence.

Line 133 – please explain “mechanical threshold”.

Based on the comment made by the reviewer #1, we deleted the sentence.

In the Chapter 2.4 only one clinical trial is mentioned, despite the title. Lines 157-9 are repetitive (written already in the lines 33-5).

Based on the comment made by the reviewer #1, we revised the title.

Line 178 – the sentence is not finished.

Thank you very much for the useful comment. We revised this sentence.

Line 210 – the reference is missing.

The reference is same reference which described above. We revised sentence.

Line 221 – this sentence needs re-phrasing, “utility preparation” is a strange term.

Thank you very much for the useful comment. We revised this sentence.

Line 229 – reference 23 is wrong and should be corrected.

We deleted the reference.

Line 253 – “the decrease in respiratory capacity was inhibited” should be re-written.

Thank you very much for the useful comment. We revised this sentence.

Line 260 – “refrigerated – impairment” should be changed.

Thank you very much for the useful comment. We revised this sentence.

Lines 293-5 – after the “although the release of cyt c…” is not clear, please explain better.

We revised this sentence in the revised version.

Line 337 – What does “Culler” stand for?

Thank you very much for the useful comment. We changed “Culler” into “Cellular”.

Line 349 – ROS production?

Thank you very much for the useful comment. We added “production” in the revised version.

Line 350 – replace “activated” with “increased”.

Based on the comment, we revised.

Line 366 – delete “system” and please explain “cellular activation”.

Based on the comment, we revised the sentence.

Line 384 – “There are an extremely wide variety of therapies targeting mitochondria is in contrast with what is said in the Line 43: “there are… no commercially available mitochondrial drugs”.

Thank you very much for the useful comment. We revised the sentence based on the comment made by the reviewer #1.

Line 394 – Add “membrane” before “potentials”.

Based on the comment, we revised.

Lines 409-16 – Ongoing projects and authors’ activities in academia and industry should not be a part of the scientific paper.

We would like to describe these activities if the Editor gave us the permission.

Lines 418 – 420 are repetitive and should be deleted.

Based on the comment, we deleted.

Line 422 – please explain what does “The formulation of mitochondria” stand for.

Based on the comment, we revised.

Line 597 – The title of the Table 3 is wrong.

Thank you very much for the useful comment. We revised the title in the revised version.

We appreciate his/her helpful and the valuable suggestions made by reviewer #1.

Reviewer 2 Report

This review from Yamada et al. is well written and addresses the current state of knowledge in mitochondrial transplantation therapy. The authors discuss various aspects of the mitochondrial transplantation therapy including relevance to diseases conditions, immunological reactions, in vivo studies, storage conditions of isolated mitochondria and drug delivery systems. Here are my comments to strengthen this manuscript:   Major:   1) In general, I'm a bit confused with the usage of the expression/term "mitochondria as a drug" or "mitochondria as a medicinal drug". If the authors cannot provide a strong rationale for using these terms/expressions, I'd rather replace it with "mitochondrial transplantation therapy" since this review is directed towards transplantation therapy.    2) In my opinion, this review lacks a critical section on the potential mechanism of mitochondrial transplantation therapy. It is important to prime the readers with details related to mitochondrial dysfunction and how mitochondrial transplantation therapy works at a mechanistic level in various disease conditions discussed in the review.    3) It will also benefit readers if the authors could weigh in on advantages and disadvantages of various mitochondrial isolation methodologies.    Minor comments:   Line 46: authors say " it would be more practical to use allogeneic mitochondria rather than autologous mitochondria as a pharmaceutical product." However, McCully et al. study which is referred to in the beginning of the introduction section, carried out an autologous transplantation procedure. It is important to highlight this particular detail of MuCully paper in this section, so that the readers can get a historical perspective.    Line 114: replace "..in the liver in was attempted.." with "..in the liver was attempted.."   Line 273-275: This sentence is mis-aligned, please rewrite "On the other hand, when rat livers were stored in the Eurocollins solution, glucose permeated into the hepatocytes. mitochondrial expansion was observed, and complex III and IV activities were lost due to the absence of antioxidants."
  Line 278: Replace "Thus, the optimizing the preservation solution" with " Thus, the optimization of the preservation solution"   Line 299: Replace "DMSO was observed.." with " DMSO was retained.."

Author Response

To reviewer #2

       We are grateful to reviewer #2 for the critical comments and useful suggestions that have helped us to improve our paper. As indicated in the responses that follow, we have taken all these comments and suggestions into account in the revised version of our paper.

Comment:

This review from Yamada et al. is well written and addresses the current state of knowledge in mitochondrial transplantation therapy. The authors discuss various aspects of the mitochondrial transplantation therapy including relevance to diseases conditions, immunological reactions, in vivo studies, storage conditions of isolated mitochondria and drug delivery systems. Here are my comments to strengthen this manuscript:

Major:

1) In general, I'm a bit confused with the usage of the expression/term "mitochondria as a drug" or "mitochondria as a medicinal drug". If the authors cannot provide a strong rationale for using these terms/expressions, I'd rather replace it with "mitochondrial transplantation therapy" since this review is directed towards transplantation therapy.

Thank you very much for the useful comment. We revised some descriptions in the revised version.

2) In my opinion, this review lacks a critical section on the potential mechanism of mitochondrial transplantation therapy. It is important to prime the readers with details related to mitochondrial dysfunction and how mitochondrial transplantation therapy works at a mechanistic level in various disease conditions discussed in the review.

Thank you very much for the useful comment. We added some descriptions in the revised version, as described below.

Lines 165-175 in the revised version,

“McCully et al. presumed that the mechanism by which transplanted mitochondria exert a protective effect on the ischemic heart is that the mitochondria promote enhanced myocardial function as described below [22]. One possibility is that transplanted mitochondria increase the ATP content and activate ATP synthesis in the cells of the heart. Another possibility is that transplanted mitochondria migrate into the heart cells by actin-dependent endocytosis, and the mitochondria then release cardioprotective cytokines. These cytokines could promote cell growth and proliferation, thus promoting angiogenesis and protecting cardiomyocytes from apoptosis. The third possibility is that transplanted mitochondria replace damaged mtDNA with normal mtDNA during ischemia. Replacing damaged mtDNA with normal mtDNA by mitochondrial transplantation could increase ATP synthesis capacity. Based on these possibilities, it is suggested that mitochondrial transplantation could increase myocardial function.” was changed

3) It will also benefit readers if the authors could weigh in on advantages and disadvantages of various mitochondrial isolation methodologies.

Thank you very much for the useful comment. Unfortunately, we were not able to add this information by the deadline.

Minor comments:

Line 46: authors say " it would be more practical to use allogeneic mitochondria rather than autologous mitochondria as a pharmaceutical product." However, McCully et al. study which is referred to in the beginning of the introduction section, carried out an autologous transplantation procedure. It is important to highlight this particular detail of MuCully paper in this section, so that the readers can get a historical perspective.

Thank you very much for the useful comment. We added some descriptions in the revised version, as described below.

Lines 35-38 in the revised version,

“They performed the autologous mitochondrial transplantation for myocardial ischemia-reperfusion injury of pediatric patients who required extracorporeal membrane oxygenation (ECMO), and recovered the most of their cardiac function and successfully separated from ECMO support.” was changed

Lines 50-52 in the revised version,

“Because there would be more suitable situation for allogenic mitochondria than autologous, including the patients of mitochondrial disease whose cells and tissues have the mutation of mitochondrial DNA and mitochondrial dysfunction.” was changed

Line 114: replace "..in the liver in was attempted.." with "..in the liver was attempted.."  

Based on the comment, we revised.

Line 273-275: This sentence is mis-aligned, please rewrite "On the other hand, when rat livers were stored in the Eurocollins solution, glucose permeated into the hepatocytes. mitochondrial expansion was observed, and complex III and IV activities were lost due to the absence of antioxidants."

Thank you very much for the useful comment. We revised this sentence in the revised version.

Line 278: Replace "Thus, the optimizing the preservation solution" with " Thus, the optimization of the preservation solution"

Based on the comment, we revised.

Line 299: Replace "DMSO was observed.." with " DMSO was retained.."

Based on the comment, we revised.

We appreciate his/her helpful and the valuable suggestions made by reviewer #2.